# How Single Amino Acid Substitutions Can Disrupt a Protein Hetero-Dimer Interface: Computational and Experimental Studies of the LigAB Dioxygenase from *Sphingobium* sp. Strain SYK-6

**DOI:** 10.3390/ijms24076319

**Published:** 2023-03-28

**Authors:** Angelika Rafalowski, Bakar A. Hassan, Kate Lou, Minh Chau Nguyen, Erika A. Taylor

**Affiliations:** Department of Chemistry, Wesleyan University, Middletown, CT 06459, USA

**Keywords:** lignin, biofuels, dioxygenase, extradiol, dimer, phenylalanine, protocatechuate, PCA, biomass catabolism

## Abstract

Protocatechuate 4,5-dioxygenase (LigAB) is a heterodimeric enzyme that catalyzes the dioxygenation of multiple lignin derived aromatic compounds. The active site of LigAB is at the heterodimeric interface, with specificity conferred by the alpha subunit and catalytic residues contributed by the beta subunit. Previous research has indicated that the phenylalanine at the 103 position of the alpha subunit (F103α) controls selectivity for the C5 position of the aromatic substrates, and mutations of this residue can enhance the rate of catalysis for substrates with larger functional groups at this position. While several of the mutations to this position (Valine, V; Threonine, T; Leucine, L; and Histidine, H) were catalytically active, other mutations (Alanine, A; and Serine, S) were found to have reduced dimer interface affinity, leading to challenges in copurifing the catalytically active enzyme complex under high salt conditions. In this study, we aimed to experimentally and computationally interrogate residues at the dimer interface to discern the importance of position 103α for maintaining the integrity of the heterodimer. Molecular dynamic simulations and electrophoretic mobility assays revealed a preference for nonpolar/aromatic amino acids in this position, suggesting that while substitutions to polar amino acids may produce a dioxygenase with a useful substrate utilization profile, those considerations may be off-set by potential destabilization of the catalytically active oligomer. Understanding the dimerization of LigAB provides insight into the multimeric proteins within the largely uncharacterized superfamily and characteristics to consider when engineering proteins that can degrade lignin efficiently. These results shed light on the challenges associated with engineering proteins for broader substrate specificity.

## 1. Introduction

The aromatic heteropolymer lignin accounts for 10–35% of lignocellulosic biomass, making it the second most abundant renewable organic material in the biosphere, after cellulose [1]. Production of fuels and fine chemicals from lignin has the potential for high sustainability and low environmental costs compared to other carbon mass sources [2]. However, effective degradation of the heterogeneous aromatic structure and conversion of lignin-derived aromatic compounds (LDACs) to high-value products remains a challenge for existing methods [1,3]. Therefore, there is considerable interest in microbial catabolism of lignin and LDACs, particularly the aromatic ring-cleaving dioxygenases, such as LigAB [4,5,6].

LigAB, of the protocatechuate dioxygenase (PCAD-Memo) superfamily, participates in the LDAC catabolic pathway of *Sphingobium* sp. strain SYK-6 (formerly *Sphingomonas paucimobilis* sp. strain SYK-6; Figure 1A) [7,8,9,10,11]. LigAB is an extradiol dioxygenase that inserts a dioxygen molecule across the C4-C5 bond of protocatechuic acid (3,4-dihydroxybenzoic acid or PCA). This reaction opens the aromatic ring to form 4-carboxy-2-hydroxymuconate-6-semialdehyde (CHMS), which is processed into central metabolic pathway molecules [12,13,14] (Figure 1B). Although many extradiol dioxygenases have the capability to degrade multiple LDACs within their corresponding pathway, LigAB has the broadest substrate utilization profile of the dioxygenases described in the literature to date [13,15]. Previously in our lab, we identified a library of compounds that function as substrates, inhibitors, or activators of LigAB [13,15,16,17]. Numerous mutants of the LigA subunit (at the 103 position) were found to enhance the catalytic activity of the LigAB complex for dioxygenation of gallate and 3-O-methylgallate (3MGA)—some mutants even outperformed the native 3MGA dioxygenase, DesZ. Understanding LigAB’s substrate specificity, tolerance of mutagenesis, and kinetic profile for its substrates would enable the development of engineered organisms that efficiently catabolize lignin into biofuels or fine chemicals.

Structurally, LigAB is a homodimer of α/β heterodimers (Figure 2A) [10]. The iron-binding active site is located at the interface between the small alpha and the large beta subunits of each dimer (Figure 2B and Appendix A). Interestingly, all but one catalytic residue, including those of the iron-binding motifs, are contributed by LigB subunit (the β domain of the heterodimer). F103α, sitting at the interface of the allosteric pocket and the active site (Figure 2B and Appendix A), is contributed by the LigA (alpha) subunit [10]. Crystallography data indicate that F103α is not involved in metal coordination or acid/base catalysis. Previous mutagenesis studies of LigAB from our lab revealed that the residue controls the enzyme’s substrate specificity through interaction with the C5-functionality of bound substrates [18]. F103A and F103S mutations of LigA were also shown to prevent the LigA and LigB proteins from copurifying. This suggested an additional role for F103α in protein–protein interactions in the LigAB complex (Figure 2B and Appendix A).

Phenylalanine plays an important role in the thermodynamic stability of interfaces in many proteins [19,20,21]. For instance, in signal transducer and activator of transcription 5 (STAT), phenylalanine at position 706 facilitates the homodimerization of the protein through forming an intramolecular network of hydrophobic interactions with other nonpolar residues at the cognate domain of the same dimer [19,20,21]. The resultant hydrophobic interface substantially contributes to the recognition of the phenylalanine and its associated hydrophobic network on the other dimer. In other systems, such as ErbB2, a transmembrane protein, phenylalanine residues in the monomers associate first to assist in the dimer formation, then rotate outwards in order that the helices can align [22]. Overall, it has been proposed that phenylalanine residues contribute to the dimerization propensity of proteins [22,23]. In the case of LigAB heterodimer, the single catalytic phenylalanine may have been evolutionarily selected to enhance the protein subunits’ ability to dimerize.

In an effort to elucidate the role of F103α in LigAB stability, we further characterized a collection of mutants that were previously studied by Barry et al. [18] with both computational and experimental methods. Quantifying and comparing the percentage of each subunit that copurifies for the mutant proteins to the wild-type protein allowed for mutant induced changes to be observed. Computational protein mutagenesis and molecular dynamic simulations allowed for the calculation of thermodynamic changes to occur on the molecular level for each of the experimentally tested mutants. Molecular dynamic simulations revealed changes in free energies of the heterodimer depending upon the identity of residue 103α and enabled determination of their relative stability, which strongly correlated to the observable and significant changes on the experimentally determined, macroscale dimerization stability.

## 2. Results

### 2.1. Protein Dimer Purification and SDS-PAGE Analysis

To purify and isolate a kinetically active form of LigAB, the gene pair is co-expressed, and the dimer is purified anaerobically. Of the two proteins, only LigA has an N-terminal His6-Tag, but the dimer associates readily in solution, and thus the dimer copurifies without any extensive steps. Since we have previously observed that SDS-PAGE analysis of both aerobically and anaerobically purified LigAB are indistinguishable (Appendix A), the purification described herein was completed aerobically. The aerobic affinity chromatography process was completed in the same manner as the previously described anaerobic purification process (Appendix A).

When purifying wild-type LigAB, in the third elution step where the fractions are collected for further buffer exchange and analysis, the two proteins are expressed at approximately the same abundance. A small fraction of LigB is initially released from the resin at a slightly higher percentage than LigA. As this elution step continues, later fractions contain a greater percentage of LigA compared to LigB. Protein abundance in each fraction is assessed and fractions are collected in order that the final ratio of LigA to LigB is about 1:1 for catalytic and other characterization steps (Appendix A, Table 1).

The purification and quantification processes were repeated for the other mutants of F103α that were previously characterized (Appendix A). Interestingly, two of the three non-polar mutants (A and L) had one fraction with a greater amount of LigB in the second elution fraction compared to the third elution fraction (Appendix A). When looking at fractions E_3–1_ to E_3–5_, the percentages of the LigA are decreased by about 40% for these mutants and the percentages of LigB are decreased by about 20 or 25% for F103L and F103A, respectively. Although this is the case, all three non-polar residues were still found to have a greater ratio of β:α than wild-type when comparing E_3–1_ to E_3–5._ This suggests that these residues may be destabilizing the α-subunit or the dimer interface in some way in order that less of the protein is stably expressed and purification is reduced. However, the overall ratio of the protein subunits remains relatively consistent when compared to wild-type.

As for the purification of the polar mutants (Appendix A), the opposite trend was observed. All polar mutations allowed for the isolation of LigAB, where the α:β ratio was greater than 1. F103S purified most similarly to wild-type (Appendix A); conversely, the other two polar mutants deviated noticeably from the wild-type. The F103T mutant was observed to have a similar abundance of the α-subunit in the fractions of interest, but with a considerably lower abundance of the β-subunit leading to a large ratio of α:β (Appendix A). In the case of F103H, the expression (total protein abundance) is diminished as compared to wild-type (Appendix A). In the fractions of interest, isolated LigA was slightly reduced as compared to WT-LigAB (<10%), whereas the expression of LigB for the F103H mutant was approximately half of those for wild-type. This suggests that although the α-subunit itself may not be destabilized greatly due to the mutation, the dimer interface may be weakened.

### 2.2. Native Gel Analysis

A native gel was carried out to understand the effect of the mutations on the formation of the catalytically active oligomerization state of LigAB (Figure 3). Due to the effective purification of F103S and F103A mutants described above, they were used as markers for their group. LigA is the smaller of the two dimers with a molecular weight of 17.711 kDa (when expressed with His_6_-tag), and LigB has a molecular weight of 33.292 kDa. Interestingly, the wild-type protein exists at a molecular weight corresponding to a trimer of dimers, although some other higher order oligomeric states exist. As seen with our denaturing SDS-PAGE analysis, the F103S variant behaves similarly to wild-type. This mutant, similar to wild-type, exists primarily as an α_3_β_3_ dimer, but there is an increase in the abundance of the higher-order oligomers (which appear to be a hexamer of heterodimers). Conversely, F103A exists more as the heterodimer (α_1_β_1_) or as a higher-order oligomer. Additionally, the bands are less distinct, suggesting perhaps that other interactions are being observed. Perhaps this can help explain the excess of the beta subunit observed for the F103A, F103V, and F103L mutants. The bands for the F103A mutant enzyme are consistent with a mass of the α_1_β_1_ heterodimer at low concentrations, but the protein band appears more elongated and smeared at higher concentrations. It may be possible that the overexpression of the hydrophobic mutants allows for the formation of other intermolecular protein interactions and warrants further study.

### 2.3. Heterodimer Stability Calculation

The free energy of LigAB heterodimer wild-type and several F103α mutant variants were calculated to determine changes in stability of the complex via thermodynamic cycle (Figure 4A). During the simulation, the wild-type and mutant variants equilibrated with an overall root mean square deviation (RMSD) of approximately less than 3.5 Å (Appendix A). This demonstrates that the simulated complexes have reached equilibrium and without dramatic variability across the trajectory, these structures were suitable for further simulations to calculate the free energy difference of the mutation relative to wild-type. We observed that the overall fluctuations of the individual residues do not vary drastically when calculating the C_α_ root mean square fluctuations (C_α_RSMF) for the wild-type and mutant proteins (Appendix A). Locally, the greatest changes to per residue fluctuations occur downstream of the F103α residue in the mutant simulations, whereas globally, residues in chain B experience greater fluctuations due to these mutations (Appendix A). The F103A, F103V, and F103L variants had a free energy difference relative to the wild-type of 1.96 ± 0.17, −0.22 ± 0.17, and −0.22 ± 0.11 kcal/mol, respectively (Figure 4B). The F103S and F103T mutants have relative free energy differences of 3.77 ± 0.20 and 0.9 ± 0.30 kcal/mol, respectively. Since the histidine side chain has two nitrogen atoms that can harbor a proton in the neutral state; therefore, we calculated free energy differences for both the delta (HID) and epsilon (HIE) form, which provided values of −0.38 ± 0.11 and 1.37 ± 0.07 kcal/mol for their free energy difference relative to the wild-type. Furthermore, the solvent accessible surface area of the entire LigAB complex was calculated, and the difference (ΔSASA; Appendix A) was determined relative to the wild-type. The serine mutation caused the burial of several other residues in the protein (Appendix A). Threonine has the opposite effect where globally, it drives other residues to become more solvent exposed. These reorganizations are relatively subtle and do not cause changes in the radius of gyration, but they could contribute to the solvation of the protein–protein interface. The other variants have negligible effects on the global desolvation of other residues. All variants at the mutated residue are more solvent exposed than phenylalanine, except for the histidine variants (Appendix A). The histidine protonated at the delta position (HID) is more buried, whereas the histidine protonated at the epsilon position does not display any large differences.

## 3. Discussion

The mutants of interest were previously identified as kinetically active mutants, F103V/T/L/H, in addition to two mutants that were catalytically inactive F103A/S^18^. We hypothesized that if the affinity of the heterodimer interface was reduced, then an excess of the α-subunit should elute off the column due to this subunit with an N-terminal His-tag. Additionally, since the β-subunit should not stick to the Ni-NTA affinity column on its own; therefore, it should be found in lower levels in the elution fractions. We analyzed the abundance of the LigA and LigB proteins and quantified the relative abundance of the subunits using Image Quant software.

In previous research, mutations at the F103α position to residues that were slightly smaller and still hydrophobic led to enhancements to catalysis for non-native substrates [18]. We hypothesized that these mutations enlarged the area of the active site, allowing for substrates with larger C5 groups to more easily bind in chemically competent poses within the active site. During this previous investigation, other hydrophobic mutations were also predicted to enhance the reactivity of non-native substrates by allowing for larger functional groups at the C5 position, but these mutants did not enable LigAB to efficiently copurify.

Upon purifying the mutants of interest for this study, where we utilized a lower salt concentration than previously reported, all mutants could now copurify to some extent. Specifically, to enable copurification of all of the mutants, the buffer was changed from 20 mM HEPES, 500 mM NaCl, pH 7.4 to 20 mM HEPES, 300 mM NaCl, pH 8.0. The change in buffer conditions facilitated the purification of all mutants, especially F103A and F103S mutants that previously purified as only the LigA protein. Without changing the buffer conditions, careful energetic analyses and comparison to our computational data would not be possible. We believe that the electrostatics of these purification conditions may have contributed significantly to the mutant proteins’ ability to copurify.

A novel property discovered of LigAB is its oligomerization state as determined by the native gel. From its crystal structure, LigAB was believed to exist as a homodimer of heterodimers (α_2_β_2_) [10], but the molecular weight of wild-type LigAB in the native gel is more consistent with a trimer of the heterodimers (α_3_β_3_), as shown in Figure 3. While this oligomerization state is unique amongst enzymes characterized in this class of dioxygenases, it is yet to be determined whether the extradiol dioxygenase activity is reliant on any one oligomerization state [24]. Further experiments, such as size-exclusion chromatography or analytical ultracentrifugation need to be completed to validate the results of the native gel. Additionally, the design of constructs to allow LigA and LigB to be individually purified in order that the oligomerization state can be further studied are planned. Furthermore, it would be valuable to determine whether LigAB is catalytically active in multiple oligomeric states. Continued studies of other related extradiol dioxygenase enzymes could provide insights on the dependence of oligomerization state on the catalytic activity for this superfamily.

When the experimental and computational data are taken together, both the size and polarity of the residue at the 103α impact the stability of the heterodimer interface. All nonpolar mutants tested were seen to have diminished abundances of both the α and β subunits relative to wild-type. When considering the ratios of the subunits, less of the α subunit is being isolated. This may be due to destabilization of the LigA protein alone or due to reduction in protein adopting a catalytically competent dimer interface. Although all mutants with non-polar amino acids at position 103 lead to protein with α:β ratios near 0.8 (suggesting an excess of the β subunit), the F103A mutation had a positive free energy difference indicating that only this nonpolar mutation may be destabilized in some way. Looking at the native gel, most of the enzyme exists in the monomeric state and other higher order oligomers. This mutant is not observed to exist in the predominant oligomeric form of wild-type LigAB. This indicates that while the two subunits copurify, the dimer interface may not be sufficiently stable over time. This may be why previous purifications were unsuccessful for the F103A mutant enzyme.

When Phe103 was exchanged for polar amino acids, the ratios of α:β are not consistent and do not follow any patterns based upon electronics or size. Previous studies by Barry et al. [18] have indicated that F103H and F103T are kinetically active, while F103S is not despite the observation that F103S has an α:β ratio most similar to wild-type. Substitution of polar residues for phenylalanine lead to a positive relative free energy, indicating that these polar mutations are destabilizing. Furthermore, the native gel for F103S indicated that while the trimer-of-heterodimers was formed, other higher order oligomers formed at greater abundances than wild-type. This suggests that the observation about purification and activity differences may result from destabilizing changes in the heterodimer ↔ active dimer form ↔ oligomer equilibrium.

Further evidence of heterodimer destabilization is provided by our molecular dynamic simulations. The hydrophobic residues (valine, leucine) have a small stabilizing effect, whereas the polar residues (serine, threonine) have a destabilizing effect. The hydrophobic residues are uniformly smaller than phenylalanine, and while they can pack more readily into the hydrophobic pocket of the active site, they provide a lower desolvation barrier than the wild-type phenylalanine. The alanine variant is destabilizing, which may be due to the small size and flexible nature of the residue. If phenylalanine is acting similarly to a cap for the active site, then alanine could be small enough where it would not act as an effective barrier between the active site and the solvent. Polar residues, such as serine and threonine are destabilizing since they force a global rearrangement for the enzyme to become drastically more solvent accessible (threonine) or inaccessible (serine). Serine seems to have a global effect that may be where the oligomeric destabilizing effect stems from. Interestingly, the effect of mutagenizing phenylalanine to histidine depends upon which nitrogen in the ring is protonated. When histidine is protonated at the delta position, it is stabilizing since it makes contact with several residues in LigB. When protonated at the epsilon position, histidine is in a hydrophobic pocket with no stabilizing interactions from other residues. The protonation of the histidine at the epsilon position (HIE) is more consistent with the experimental data, indicating that this may be the experimentally relevant form, and thus explaining why this mutant has the lowest α:β ratio of all the mutants.

## 4. Materials and Methods

Commercially available reagents and solvents were purchased from Fisher Scientific, apart from ampicillin, which was purchased from Goldbio, and acrylamide/bisacrylamide, which was purchased from Bio-Rad Laboratories. DH5α and BL21 (DE3) chemically competent *E. coli* cells were purchased from New England Biolabs (Ipswitch, MA, USA). Centrifugation was performed on DuPont Instruments (Wilmington, DE, USA) Sorvall RC-5B centrifuge. All cells were lysed using an Avestin (Ottawa, ON, Canada) Emulsiflex-C5 high-pressure homogenizer.

### 4.1. Molecular Dynamic Simulations and Relative Binding Free Energy Calculation

All molecular dynamic simulations were performed with the Gromacs 2022.1 package and the Amber99sb forcefield in triplicate to ensure reproducibility [25,26]. All wild-type and mutant structures were derived from the wild-type LigAB heterodimer (PDB: 1BOU) and were prepared from chain A and B. Missing loops and sidechains were modeled with Prime [27,28]. Protonation states for ionizable sidechains were determined with PROPKA [29,30]. The pmx extension was used to generate hybrid structures and topologies for the wild-type (λ = 0) and mutant (λ = 1) state [31,32]. Each system was placed into a dodecahedron periodic boundary condition with a 10 Å buffer region. The system was solvated with a TIP3P water model and electroneutralized with counterions to a final concentration of 0.150 M. One thousand steps of a steepest descent were used as energy to minimize the system followed by isochoric/isothermal (NVT) and isobaric/isothermal (NPT) equilibrations for a total of 1 ns at 298 K. Furthermore, all-atom restraints were applied during equilibration and were slowly released during an extended 500 ps (NPT) starting with sidechains, then backbone. The equilibrated system was then subject to a 50 ns simulation and a total of 200 frames, which were evenly spread across the second half of the trajectory that were subject to non-equilibrium simulation. Herein, the λ value was varied from 0 to 1 or vice versa for 200 ps with a softcore potential [33]. The free energy was determined based on Crooks fluctuation theorem [34,35]. Simulations were performed with a 2 fs timestep. Long range electrostatics were calculated with the particle-mesh-ewald with a grid spacing of 1.2 Å and a 4th order cubic interpolation. Short range nonbonded interactions are calculated with a 11 Å cutoff. Temperature and pressure are coupled with the v-rescale and Parinello-Rahman thermostat and barostat, respectively. Hydrogens were constrained with holonomic constrains through the LINCS method [36]. The solvent accessible surface area (SASA) was calculated in GROMACS based on the double cubic lattice method [37]. The SASA is defined as the surface of a sphere between the solvent radius of the probe and the Van der Waals radius of atoms in question without overlapping with other atoms. When performed on the entire protein, this defines the area that can be penetrated by a solvent molecule.

### 4.2. Protein Expression and Purification

All mutants for this study (A, S, V, L, H, T) were made via site-directed mutagenesis as previously described by Barry et al. [17]. An overnight culture of *E. coli* BL21 cells containing a pET15b plasmid containing the genes for LigA and LigB (wild-type or mutant version) was inoculated into 15 mL of Luria Broth (LB) media supplemented with ampicillin (100 μg/mL) and allowed to shake at 37 °C overnight. This seed culture was used to inoculate 2 L of LB supplemented with 100 µg/mL of ampicillin. Cells were allowed to shake at 200 rpm and 37 °C until an OD_600_ of 0.4–0.6 was reached, at which point, gene expression was induced with the addition of isopropyl β-d-thiogalactopyranoside (IPTG) to a final concentration of 1 mM. The culture was allowed to grow for an additional 24 h after induction. Cells were harvested by centrifugation at 5422× *g* for 10 min and lysed via an Avestin (Ottawa, ON, Canada) Emulsiflex-C5 high-pressure homogenization with 5–7 passes at approximately 15,000 psi. The lysate was centrifuged at 21,728× *g* for 40 min at 4 °C to pellet insoluble cellular debris. The supernatant was loaded onto 15 mL of HisPur Ni-NTA resin pre-equilibrated with bind buffer (50 mM HEPES, 300 mM NaCl, 10 mM imidazole, pH 8.0). The His-tagged LigAB enzyme was washed with wash buffer (50 mM HEPES, 300 mM NaCl, 20 mM imidazole, pH 8.0) and eluted by an imidazole step gradient. The buffers contained 50 mM HEPES, 300 mM NaCl at pH 8 with, increasing imidazole sequentially, 62.5 mM (2.5 CV), 125 mM (2.5 CV), and 250 mM (5 CV) imidazole. Fractions (5 mL) were collected for all elution steps.

### 4.3. SDS-PAGE and Analysis

One flow through fraction (FT), one wash fraction (W), every other fraction eluted with 62.5 mM imidazole (E_1_), and every fraction eluted with 125 mM and 250 mM imidazole (E_2_ and E_3_, respectively) were analyzed by SDS-PAGE (15/6% acrylamide). After electrophoresis, gels were incubated in a fixing solution (40% *v*/*v* ethanol, 10% *v*/*v* Glacial Acetic Acid) for at least 20 min. The gels were then stained using a Coomassie stain (10% acetic acid, 40% methanol, 605 µM Coomasie Brilliant Blue) for at least 20 min. The bands were resolved using a destain solution (10% acetic acid, 20% methanol). The gels were immediately digitized on a Typhoon FLA 9000 using a 473 nm laser. Image Quant Program (GE Healthcare Biosciences, Chicago, IL, USA) was utilized to analyze each mutant gel and quantify the relative amount of protein in each subunit band.

## 5. Conclusions

While we previously reported that mutations of LigAB at the F103α position have resulted in broader substrate utilization profiles, these mutations had variable impacts on the overall stability of the enzyme complex. Analysis of SDS-PAGE separations for these proteins indicates that non-polar mutants: V, L, and A, purify a greater amount of LigB when compared to wild-type. This may be due to the mutation causing a minor destabilization of LigA, which causes it to purify at lower amounts during the elution steps, or that somehow the LigB protein interacts with the LigA/B dimer even in the absence of its cognate LigA. These mutations may prevent the proteins from properly forming the correct oligomeric form; rather forming only the monomeric heterodimer complex and higher-order oligomers or versions with skewed α:β ratios. Conversely, the polar mutations, as they become bulkier, express more LigA compared to LigB. This could be due to the destabilization of the dimer interface resulting from the mutation. These mutations form oligomers observed for wild-type, but also allow for the formation of higher-order oligomers not observed for wild-type LigAB. This study provided insight into the stability and oligomeric states of a series of LigAB mutants that were previously identified to alter the substrate utilization profile of this enzyme, and caution is always warranted when attempting to reengineer protein specificity. Further studies of these LigAB mutants are planned to determine whether the subunits can be individually expressed and the active complex reconstituted, in order to enable determination of the thermodynamic parameters of dimer association. Additionally, investigation of the impact of oligomerization state(s) on protein catalysis and stability will enable understanding of the role of the observed monomers and trimers of the αβ-LigAB heterodimer, in order that future efforts for enzyme redesign will ensure high levels of enzyme activity for mutant versions that are able to catabolize a variety of substrates.

## Figures and Tables

**Figure 1 ijms-24-06319-f001:**
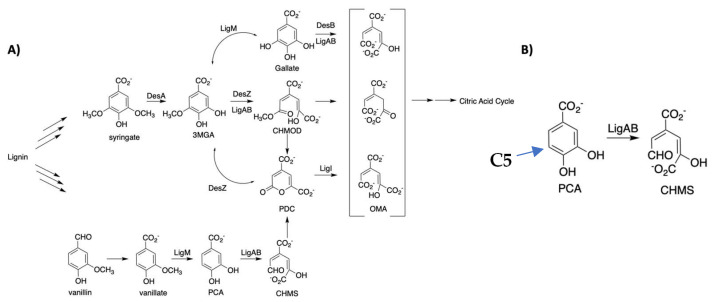
Lignin degradation pathway in which LigAB catalyzes the deoxygenation reaction of various phenolic compounds. (**A**) A portion of the lignin degradation pathway from *Sphingomonas paucimobilis* sp. strain SYK-6 in which LigAB is found. LigAB catalyzes the ring opening of its endogenous substrate, 4,5-protocatechuic acid, but also other LDACs within and outside of the shown pathway. (**B**) The aromatic ring opening reaction of protocatechuic acid, the native substrate of LigAB. The F103α recognizes functional groups on the C5 position (indicated with a blue arrow). The H at the C5 position of PCA can be substituted with hydroxyl and hydroxymethyl groups, to respectively yield gallate and 3MGA.

**Figure 2 ijms-24-06319-f002:**
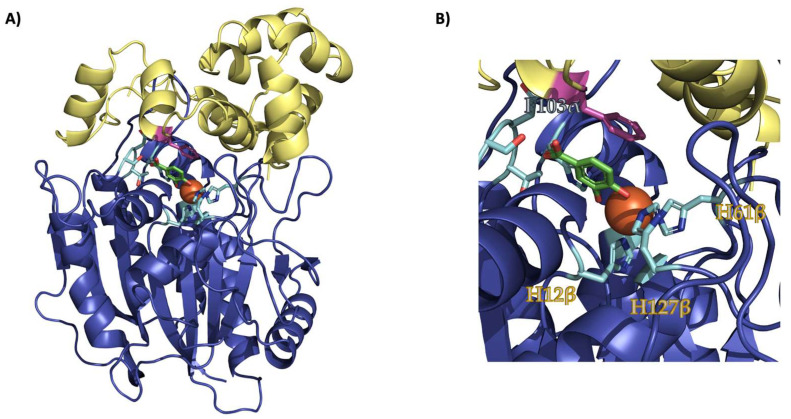
The interface complex of LigAB with Fe(II) and PCA bound. (**A**) The heterodimer LigAB (PDB: 1B4U) with LigA in yellow, LigB in blue, the active site residues in cyan, F103 in pink, PCA in green, and Fe(II) in orange. (**B**) A close-up of the active site with the active site, which illustrates the proximity of F103α to the coordinated substrate.

**Figure 3 ijms-24-06319-f003:**
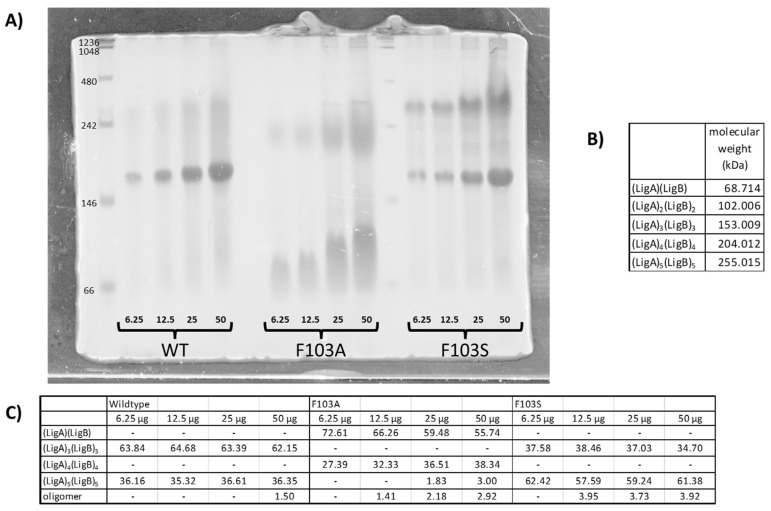
Native gel analysis of wild-type LigAB, F103A, and F103S mutants. (**A**) A 10% Native Polyacrylamide gel of wild-type, F103A, and F103S stained with Coomasie Blue. The bands show the different oligomerization states of the mutants compared to the wild-type, where protein samples are varied from 6.25 to 50 ug of protein loaded per lane. (**B**) The calculated molecular weights of the oligomerization states as seen on the Native Polyacrylamide gel. (**C**) The percentages of the mutations in the various dimerization states of LigAB and the mutants as determined by ImageQuant.

**Figure 4 ijms-24-06319-f004:**
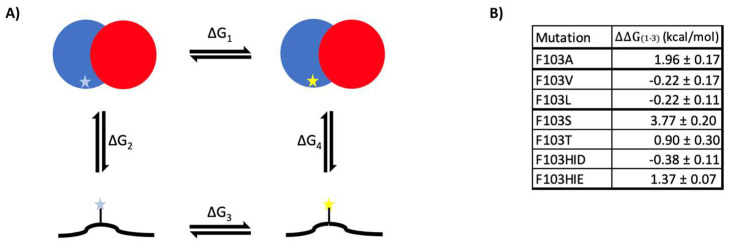
Free energy determination of F103α mutants. (**A**) Stability of LigAB was determined through determination of free energy differences between wild-type and mutant heterodimer of LigAB in folded and unfolded state based on the thermodynamic cycle, where ΔΔG_1–3_ is the free energy of the mutation. (**B**) The hydrophobic mutations (valine, leucine) are stabilizing relative to the wild-type, while polar and highly flexible mutations are strongly destabilizing (alanine, serine, histidine, threonine).

**Table 1 ijms-24-06319-t001:** The calculated abundances of the α and β subunits for wild-type LigAB and F103α mutants. The ImageQuant calculated abundance of each α and β subunit in the elution fractions E_3–1_ to E_3–5_ compared to the relative abundances of the subunit in all the elution fractions (see Appendix A). The ratios of α to β and β to α are also reported as a basis of comparison.

	α	β	α:β	β:α
Wild-type	81.1	75.3	1.077	0.928
F103A	46.8	57.3	0.817	1.224
F103V	56.7	71.8	0.790	1.266
F103L	51.7	58.8	0.879	1.137
F103S	69.0	66.2	1.042	0.959
F103T	80.7	49.1	1.644	0.608
F103H	71.1	32.4	2.194	0.456

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
