# Peer review of "How Single Amino Acid Substitutions Can Disrupt a Protein Hetero-Dimer Interface: Computational and Experimental Studies of the LigAB Dioxygenase from Sphingobium sp. Strain SYK-6"

_ijms, 2023, doi:10.3390/ijms24076319_

Round 1

Reviewer 1 Report

The manuscript entitled "How Single Amino Acid Substitutions can Disrupt a Protein Hetero-Dimer Interface: Computational and Experimental Studies of the LigAB Dioxygenase from Sphingobium sp. Strain SYK-6" by Angelika Rafalowski et al. is computational and experimental studies to investigate the impact of the mutation at a significant site (residue F103 in alpha subunit) at the interface of LigAB Dioxygenase heterodimer. Both MD simulation and electrophoretic mobility assays have been used in this study. Instead of all 19 substitutions, only six mutations at position 103 are considered (nonpolar and polar). This study demonstrates that nonpolar/aromatic amino acid mutations in this position are preferred, but they result in oligomer destabilization. While the polar mutation may produce a dioxygenase with a useful substrate utilization profile. This research could lead to better development of engineering proteins for efficient lignin degradation by providing insight into the purification of LigAB mutants. However, the MS should be major revised to address the following issues:

Major comments:

1) The authors should investigate the actual disruption networks at the interface using structure analysis (HBs, hydrophobic interaction, etc.) based on cluster structures extracted from the MD simulation.

2) In this study, the MD simulations are not long enough and lack the replicates (one MD simulation across 50 ns) that are run in most good studies. To validate the results from MD simulations, it recommends doing one extra MD run at the same condition for each case (WT and corresponding mutant case).

3) Update figure 2(A) with better representation and different color. Let's say only using the ribbon representation without the surface representation, blue and gold colors for the dimer. Similarly, update figure 2B and label each residue and iron in the active site. What is the green residue in the active site?

4) Line 152, I think the authors mean Figure 4A and this figure is missing in the MS. The same is true for Figure 3 (b) in line 160. Please correct the number and include this figure.

5) The authors should make this point more clearly why MD simulations of the other 13 substitutions at 103 position were not investigated.

6) Could the authors do the binding assays (such as ELISA and BLI) to investigate the impact of mutation at 103 position?

7)  Limitations of the work should be mentioned in the discussion or conclusion section.

8) In section 4.1, the authors should explain in more detail how they create the initial complex structure for the mutation cases.

9) Each figure should have a unique caption that includes a brief description of the figure along with details of figure (A), (B), etc.

10) Lines 335 to 394 should fill out accurately and do not leave as in the template.

Minor Comments:

1)      Line 7 and 8, there is no “1” superscript at line 7, no “#” affiliation.

2)      All table titles (main text and SM) should appear above each table, not below.

3)      Some places should be cited properly such as:

-Line 70 to 74: “For instance, in signal transducer and activator of transcription……hydrophobic network on the other dimer.”

-Line 175 to 176: “The mutants of interest were previously identified kinetically active mutants, F103V/T/L/H, in addition to two mutants that were catalytically inactive F103A/S.”

-Line 182: “….using ImageJ software.”

-Line 226 to 227: “Previous studies have indicated that F103H and F103T are kinetically active,…most similar to wild-type.”

4)      The author should carefully correct all figure numbers in both the main text and the supplementary material. There are numerous mistakes in labeling figure numbers. For example, in line 154, Figure S5 should be changed to Figure S4. Furthermore, all figures in the supplementary should be mentioned in the main text. For example, Figure S3 does not be listed in the main text as well as Figure S2 B, C, D, F. Figures of SM should be mentioned and presented in numerical order. For example, Figure S7C in line 129 is missing and mentioned before Figure S5.

5)      The authors should use either a one-letter or a three-letter code for amino acids throughout the entire manuscript.

6)      Explain how SASA is estimated?

7)      Correct all typos.

-Line 67: “Phe103a” to Phe103α.

-Line 135: “native” to Native.

-Line 162: “two nitrogen molecules” to two nitrogen atoms

Reviewer 2 Report

In this report, the authors performed biochemical and computational studies of LigAB dioxygenase from Sphingobium sp. SYK-6. This paper investigated impact of a single amino acid substitution at position Phe103 of the alpha subunit (LigA) for the hetero-dimer formation. However, this paper only provided the biochemical denaturing SDS-PAGE and native-PAGE analyses in the affinity chromatography for the evaluation. Thus, the paper’s claim is too speculative at this stage. In addition, the correlation analysis between experimental and theorical studies was insufficient. In view of this critical point, this reviewer would not recommend the manuscript for publication in International Journal of Molecular Sciences. 

Major points:

Table 1: Although this reviewer could completely understand the correlative results, not understand how to calculate the values of alpha and beta since definition of 100% was not stated. This reviewer finds this assay method problematic. The results of F103A, F103V and F103L indicated that excess amount of beta-subunit existed in the elute fraction. Therefore, three binding modes are considered: (1) beta-subunit interacted with non-canonical site of alpha-subunit; (2) with beta-subunit other than the alpha-subunit interface; (3) with Ni-NTA resin non-specifically. Thus, this reviewer is afraid that only this assay data will not be able to evaluate the hetero-dimer formation accurately. 

Page 5, lines 152 and 160: This paper described free energy of wild-type and mutant LigAB hetero-dimer. However, the figure number was misassigned (Fig. 3A and B) and no figure regarding this data was represented. 

Page 5, lines 153-154: It did not state what can be said from the FigS4 RMSD data.

Figure S5: Mapping of residues showing higher RMSF values on the LigAB crystal structures during the MD simulation would be informative. This paper mentioned that Ala, Val, and Leu in the alpha-subunit and Ser, Thr and His in the beta-subunit showed significant fluctuations as compared with wild type. However, since positions of each amino acid residue were not indicated, detail of the dissociation mechanism of the LigAB hetero-dimer is unclear. 

Minor points:

Fig. 2: Position of alpha (blue) and beta (green) subunits should be labeled. 

Page 4, line 134: “Native Gel A” should read as “Native Gel Analysis”

Page 4, line 136: “alpha2beta2 dimer of heterodimers” should read as “alpha3beta3 dimer of heterotrimers” according to the native-PAGE data (Fig. 3, around 150 kDa). 

Figure 3A: Labels on SDS-PAGE are very difficult to read. 

Page 5, line 154: “Fig S5” should read as “Fig S4”

Fig S2: Labels (square and value) on SDS-PAGE image should be removed. 

Reviewer 3 Report

Rafalowski A. et al study protein hetero-dimer interfaces and they analysis single amino acid effects. The study is based on experimantal part by using SDS-PAGE analysis and computational with  Gromacs MD package. Authors explains about dimerizaations of proteins in introduction. The results are seprated into 3 parts: Protein Dimer Purification and SDS-PAGE analysis; Native Gel A 134; Heterodimer Stability Calculation. Then the follows discusions about heterodimers and importance of histidine. However, it is very dificult to follow the study and it should be clearified:

1. In discusion I would like to find ideas what can we exepect if histidine would be changes in other lygans. Should results be the same or ho they should corolate with the findings?

2.Line  14.  The aabstract starts with the „C5 position“. The Introduction must have more detailed explanation with references where I could find about the positions. As I understand it is one of the important part the autors study.

3. Lines: 90, 91,110, 147. Captions of Figures and Tables are as figures. They must be as text without red lines (showing errors in text).

4. Figure 3. Must be splited into Figure and Tables as Figure B and C are tables!

5.Line 67. „Phe103a“ must have reference in order to follow the text (e.g. figures in suplementary)

6. Line 162. „Histidine“ must be explained in more details and referenced to the paticular histidine.  Is it true the protein has only one histidine? The same applies to disuscution.

7.Lines 335-395 must re revised as there are not information, e.g. author contributions, declarations etc.

8. Line 53. It is unclear what does it mean „3OMG“

9. Line 167 must have citation for PDB structure.

10. As understan the mutations were made artificialy. Thus I would expect the final structures would be added as suplemnetary data.  

Round 2

Reviewer 1 Report

The authors have satisfactorily addressed most of my concerns, and the revised version of the manuscript is improved significantly. Therefore, I can recommend publication after minor modifications:

1)      Figure captions should be modified with more informative captions in both the main text and the Supplementary. For example, in figure S1, use “Active site of the interface LigAB complex with mutations at 103α residue" instead of "Active site with mutations". In Figure 2, use “The interface complex of LigAB with Fe(II) and PCA bound” instead of “Pymol images of LigAB with Fe(II) and PCA bound”. The captions of Figures 3 and 4 should also be modified.

2)      Short sentences should be used instead of long sentences throughout the manuscript, so rewrite these sentences at lines 63, 119, 122, 141, 145, 170, and 172.

3)      Typo at Line 209: change “Figure S8A” to Figure S9A. 

Author Response

Reviewer 1

The authors have satisfactorily addressed most of my concerns, and the revised version of the manuscript is improved significantly. Therefore, I can recommend publication after minor modifications:

1)      Figure captions should be modified with more informative captions in both the main text and the Supplementary. For example, in figure S1, use “Active site of the interface LigAB complex with mutations at 103α residue" instead of "Active site with mutations". In Figure 2, use “The interface complex of LigAB with Fe(II) and PCA bound” instead of “Pymol images of LigAB with Fe(II) and PCA bound”. The captions of Figures 3 and 4 should also be modified.

The authors have edited all figure legends, so they are more descriptive.

2)      Short sentences should be used instead of long sentences throughout the manuscript, so rewrite these sentences at lines 63, 119, 122, 141, 145, 170, and 172.

The authors edited the lines listed so that they are shorter and clearer for the reader.

3)      Typo at Line 209: change “Figure S8A” to Figure S9A. 

The authors have corrected the typo to reflect the correct figure- Figure S7 in line 209, and other lines.

Reviewer 2 Report

Since no additional experiment has been performed in the revised manuscript, the validity of the paper's claims remains weak. However, the authors adequately analyzed and discussed the currently obtained results. This reviewer would recommend this manuscript for publication in the International Journal of Molecular Sciences. Fig 3A still difficult to see the labels. In contrast, Figs S2-S5 are easily readable. This is an editorial issue and this reviewer leaves it to the editor's decision.

Author Response

The authors have changed the formatting and in image labeling of the figure to improve readability.

Reviewer 3 Report

 it can be published with some changes:

The authors didn't updated manuscript and some tables are as figures:

Figure 3 should be split into Figure 3 and Table 2, Table 3. The descriptions must be provided.

Figure 4 should be split into Figure 4 and Table 4. The descriptions must be provided.

All tables (Table1 and in Figure3, 4) must have the same style and font sizes.

Line 345 has reference by direct doi  link. It must be corrected.

Lines 60,63 and 69 has text with yellow color. It is unclear why it is left.

Author Response

Reviewer 3

 it can be published with some changes:

The authors didn't updated manuscript and some tables are as figures:

Figure 3 should be split into Figure 3 and Table 2, Table 3. The descriptions must be provided.

Figure 4 should be split into Figure 4 and Table 4. The descriptions must be provided.

The authors thank the reviewer for this suggestion. The authors however believe that the manuscript will be clearer if the tables and figures are not split. The authors have thus kept the previous formatting.

All tables (Table1 and in Figure3, 4) must have the same style and font sizes.

The authors changed the table in Figure 4 to match the other tables in figures.

Line 345 has reference by direct doi  link. It must be corrected.

The authors have removed the DOI and added the information as a reference.

Lines 60,63 and 69 has text with yellow color. It is unclear why it is left.

The authors removed the highlighting within the manuscript.